# Gait Reconstruction Strategy Using Botulinum Toxin Therapy Combined with Rehabilitation

**DOI:** 10.3390/toxins16070323

**Published:** 2024-07-19

**Authors:** Takatoshi Hara, Toru Takekawa, Masahiro Abo

**Affiliations:** 1Department of Physical Rehabilitation, National Center Hospital, National Center of Neurology and Psychiatry, Tokyo 187-8551, Japan; 2Department of Rehabilitation Medicine, The Jikei University School of Medicine, Tokyo 105-8461, Japan; bamboo@apricot.ocn.ne.jp (T.T.); abo@jikei.ac.jp (M.A.)

**Keywords:** botulinum toxin A therapy, spasticity, rehabilitation, lower limbs, motor function, gait

## Abstract

Numerous studies have established a robust body of evidence for botulinum toxin A (BoNT-A) therapy as a treatment for upper motor neuron syndrome. These studies demonstrated improvements in spasticity, range of joint motion, and pain reduction. However, there are few studies that have focused on improvement of paralysis or functional enhancement as the primary outcome. This paper discusses the multifaceted aspects of spasticity assessment, administration, and rehabilitation with the goal of optimising the effects of BoNT-A on lower-limb spasticity and achieving functional improvement and gait reconstruction. This paper extracts studies on BoNT-A and rehabilitation for the lower limbs and provides new knowledge obtained from them. From these discussion,, key points in a walking reconstruction strategy through the combined use of BoNT-A and rehabilitation include: (1) injection techniques based on the identification of appropriate muscles through proper evaluation; (2) combined with rehabilitation; (3) effective spasticity control; (4) improvement in ankle joint range of motion; (5) promotion of a forward gait pattern; (6) adjustment of orthotics; and (7) maintenance of the effects through frequent BoNT-A administration. Based on these key points, the degree of muscle fibrosis and preintervention walking speed may serve as indicators for treatment strategies. With the accumulation of recent studies, a study focusing on walking functions is needed. As a result, it is suggested that BoNT-A treatment for lower limb spasticity should be established not just as a treatment for spasticity but also as a therapeutic strategy in the field of neurorehabilitation aimed at improving walking function.

## 1. Introduction

Patients after stroke hemiplegia may exhibit symptoms of spasticity, a hallmark of upper motor neuron syndrome [1]. Previous reports have shown that spasticity is present in 19% of patients at 3 months and 38% of patients at 12 months after stroke [2,3]. Spasticity may hinder rehabilitation and potentially impede improvements in activities of daily living (ADL) and social reintegration due to continuous spasticity, leading to muscle atrophy, joint contractures, and pain caused by shortening of muscle fibres and ligaments [4]. Additionally, spasticity in the lower limbs can restrict joint range of motion (ROM) necessary for normal walking and regulate muscle tension [5]. Lower limb spasticity may lead to sustained increased tension in the calf muscles, potentially causing equinus [6]. An equine foot can result in ankle instability during the weight-bearing phase and poor toe clearance during the swing phase of walking [7].

Botulinum toxin (BoNT) is produced by *Clostridium botulinum*, and is the causative agent of botulism, a rare but serious disease in animals and humans [8,9]. However, the biological and toxicological characteristics of BoNT make it a valuable tool for studying neurophysiology and developing effective therapies for various human diseases [9]. Botulinum toxin is classified into seven serotypes (BoNT-A to G), with type A used clinically for spasticity [10]. Botulinum toxin type A (BoNT-A) therapy temporarily reduces muscle activity by preventing the release of acetylcholine upon neuromuscular injection, leading to reduced spasticity and muscle tone [11]. Pharmacologically, the effects of intramuscular injection of BoNT-A begin 2–4 days after injection, with peak efficacy expected at three weeks [12]. Previously reported randomized controlled trials (RCTs) have demonstrated that BoNT-A therapy reduces spasticity [13,14]. Therefore, the effectiveness of BoNT-A for spasticity is well-established with strong supporting evidence as a treatment for upper motor neuron syndrome. The American Academy of Neurology recommends BoNT-A as class A for the treatment of spasticity in adults by addressing spasticity, joint ROM, hygiene, and pain [15]. However, in the 2008 report, there were few reports of functional improvement. Despite the approval of BoNT-A in various countries over a decade ago, the approach to treating spasticity has changed dramatically. It is crucial to explore avenues to maximise the effectiveness of BoNT-A for lower-limb spasticity, promote functional improvement, and contribute to gait rehabilitation. This paper discusses the multifaceted aspects of spasticity assessment, administration, and rehabilitation with the goal of optimising the effects of BoNT-A on lower-limb spasticity and achieving functional improvement and gait reconstruction.

## 2. Evaluation of Spasticity

There are various subjective and objective methods to assess spasticity, each with potential advantages and disadvantages. Assessment of spasticity should be based on these understandings. Spasticity is considered a positive sign of upper motor neuron syndrome. Lance’s widely recognised definition characterizes spasticity as a “motor disorder characterized by velocity-dependent increase in tonic stretch reflexes, including the presence of tendon reflexes, as a component of upper motor neuron syndrome” [16]. Although the initial definition emphasises tendon reflexes in an abstract manner, a later definition proposed by Pandyan et al. states that “spasticity is a sensorimotor control disorder resulting from damage to the upper motor neuron, manifested as intermittent or sustained involuntary activation of muscles”, highlighting its pathophysiology as a consequence of central control impairment rather than peripheral limb tension [17]. Past reports on how spasticity was defined indicate that among 250 papers, 35% used the term “muscle tone”, 31% used the Lance’s definition, and the remaining percentage remained undefined [18]. Therefore, even though many researchers involved in spasticity have a common understanding of the existence of a pathology called spasticity, it can be recognized that there are various ways of expressing the common definition of spasticity. The assessment of spasticity in clinical practice commonly employs the modified Ashworth Scale (MAS). In our previous systematic review of BoNT-A and rehabilitation, the MAS was the most commonly used evaluation for spasticity [19]. It is considered superior in clinical applications, as it allows straightforward evaluation with the same indicator across all body parts. However, there are negative opinions regarding the use of MAS. Balci et al. reported that spasticity is velocity-dependent; therefore, changing stretch velocity may alter MAS measurements [20]. Fleuren et al. also argued that passive movements in assessment must consider reflex muscle activity and non-neurological characteristics and should account for changes in the elastic-viscoelastic properties of joint structures and soft tissues after upper motor neuron lesions [21]. Therefore, MAS Scale 3 was described as “considerable increase in muscle tone, passive movement difficult”. The interpretation of this difficulty in passive movement is crucial; however, in cases of typical severe spasticity, factors, such as disuse atrophy of the muscles and changes in the surrounding elastic-viscoelastic properties, must be considered. However, the MAS evaluation makes it challenging to distinguish the weight of pure spasticity from that of other contributing factors.

The modified Tardieu Scale (MTS) was used to measure joint ROM and quality of muscle reactivity. Muscle stretching was defined at three speeds: V1 (as slow as possible), V2 (falling under gravity), and V3 (as fast as possible). The ROM is determined by stretching the muscle at V3 speed, and the angle at which the initial “catch” occurs is defined as R1. The maximum ROM when stretched at V1 is defined as R2. The difference between R2 and R1 (R2 − R1) reflects the contribution of elastic and viscoelastic properties of soft tissues constituting a joint when small, and mainly reflects reflexive elements due to the stretching reflex when large [22]. Therefore, compared with the MAS, the MTS may reflect the viscoelastic properties of the structures around the joint. However, reviews on the MTS suggest that although its use is not discouraged based on reliability and validity, the evidence is insufficient [23]. Lou et al. described the MAS and MTS as subjective evaluations in the literature on spasticity assessments [24]. Clinically, these assessments are considered sufficient for evaluating spasticity and measuring its changes; however, additional evaluations may enhance the precision of spasticity treatment. As supplementary measures, Lou et al. presented nerve conduction tests, surface electromyography, and peripheral evaluations, such as NeuroFlexor [25], Myotonometer [26], and Sonoelastography [27]. These non-invasive methods allow for the quantitative evaluation of spasticity. With the increasing use of ultrasonography during BoNT-A injections in recent years, muscle evaluation has become feasible. Therefore, Sonoelastography has attracted considerable attention in recent years.

From a different perspective, evaluation using three-dimensional motion analysis may be useful, as it allows the simultaneous assessment of both spasticity and functional improvements. Tanikawa et al. evaluated toe-pointing during walking before and after BoNT-A injection; they reported the most significant improvement at six weeks post-injection [7]. Therefore, it is recommended to assess not only spasticity at rest, but also during movements, especially in patients who experience inward rotation during activities or walking. Although detailed evaluations take time, they may contribute to the establishment of a new direction for muscle selection based on surface electromyography evaluations in the future.

Proper evaluation of spasticity is crucial for selecting appropriate injection sites and doses, making it a key factor in improving function after BoNT-A injection.

## 3. Muscle Fibrosis and Sonoelastography

In this section, we discuss fibrosis associated with spasticity, its evaluation, and recent new ways to deal with it. In patients presenting with severe upper and lower limb spasticity in the chronic phase of stroke, immobility due to prolonged paralysis leads to secondary impairments, such as joint contractures and muscle fibrosis [28]. However, it remains unclear whether fibrosis occurs due to paralysis and immobility associated with spasticity, or whether it occurs due to disuse and immobility, leading to subsequent spasticity. The relationship between spasticity and fibrosis and the underlying mechanism are poorly understood. Two previous studies investigated the relationship between the efficacy of BoNT-A and degree of fibrosis [29,30]. Both studies used ultrasound echo to evaluate muscle fibrosis using the Heckmattt scale (HMS), which assesses fibrosis based on the brightness of the muscles and bones. Picelli et al. injected 500 U into the inner and outer sides of the gastrocnemius muscle, evaluated spasticity and HMS before and four weeks after injection, and observed significant improvements in MAS, Tradieu scale, and ankle ROM in all patients. However, when analysed using the HMS, patients with high values (indicating more severe fibrosis) showed less improvement. In the HMS III and IV, no significant improvements were observed in spasticity or ankle ROM. Hara et al. conducted a similar verification using intensive rehabilitation during hospitalisation [30]. In a study involving 102 patients with stroke, HMS of the gastrocnemius muscle was assessed by ultrasonography before BoNT-A injection and MAS scores of the knee and ankle joints were significantly reduced in all groups. Additionally, significant improvements in ankle ROM were observed in the HMS I–III groups, whereas no improvement was observed in the HMS IV group (indicating the most severe fibrosis). Evaluation of walking ability and balance using the 10 m walking speed, Functional Reach Test (FRT), and Timed Up and Go Test (TUG) showed significant improvements in 10 m walking speed and FRT in the HMS II and III groups, and significant improvement in TUG in the HMS I–III groups. Both Picelli and Hara’s studies showed no improvement was observed in the HMS IV group, indicating that even with the addition of rehabilitation, sufficient efficacy of BoNT-A may not be achieved in muscles with advanced fibrosis [29,30].

Considering these factors, evaluations using sonoelastography, as mentioned earlier, play a significant role. Previous reports have focused on BoNT-A and sonoelastography of the upper limbs, with studies evaluating changes in the biceps brachii before and after BoNT-A administration. These studies used elastography to calculate strain ratios and reported a significant correlation between MAS changes and elastography changes before and after BoNT-A therapy [31,32]. Although reports on the lower limbs are not common, a pilot study evaluating the elasticity of muscles and the Achilles tendon found no significant changes [33]. Hasegawa et al. evaluated shear wave velocity (SWV) using BoNT-A and performed rehabilitation, and found no significant change in resting SWV of the lower limb muscles after approximately one month of treatment [34]. However, a significant change in SWV at maximum ankle dorsiflexion was observed before and after BoNT-A therapy and rehabilitation intervention. Although SWV did not correlate with HMS or MAS, a significant correlation was found between “R2 before intervention” and SWV in groups with changes in R2 based on MTS. Thus, SWV may be used as an auxiliary evaluation tool for spasticity in cases where it is difficult to assess the effects of BoNT-A, especially in muscles with advanced fibrosis. The SWV may provide new insights into spasticity; however, further studies are required.

In the context of addressing advanced fibrosis, Extracorporeal Shock Wave Therapy (ESWT) may be effective for treating advanced fibrosis. Extracorporeal Shock Wave Therapy has gained attention in recent years as a treatment for spasticity, and several hypotheses have been proposed regarding its mechanisms: (1) involvement in the induction of nitric oxide synthesis for the formation of new neuromuscular junctions; (2) decreased excitability of motoneurons due to continuous or intermittent pressure on tendons by ESWT; and (3) decrease in acetylcholine receptors at neuromuscular junctions and inhibition of neuromuscular transmission due to ESWT [35]. Therefore, ESWT may be effective for treating spasticity with advanced fibrosis. Systematic reviews of ESWT for spasticity in the upper and lower limbs have reported improvement in all studies without apparent adverse events [35,36]. Some studies have reported improvements in walking speed and dynamic balance. Hsu’s network meta-analysis comparing the effects of ESWT and BoNT-A on spasticity, including that in the upper limbs, suggested that ESWT is equal to or more effective than BoNT-A. However, there have been no comparative studies of BoNT-A and ESWT in the lower limbs. Additionally, even in these systematic reviews, it was difficult to make a general comparison because of the differences in BoNT-A dosage and administration site. Furthermore, the mechanisms of action of BoNT-A and ESWT differ, as demonstrated by studies showing no difference in efficacy between the motor and non-motor points of ESWT. There was no difference in effectiveness between the motor and non-motor points of the ESWT. Therefore, it is suggested that the mechanism of action on spasticity may be different between ESWT and BoNT-A. Thus, we believe that there is room for debate in conducting a network meta-analysis to examine the effects of BoNT-A and ESWT as the same treatment technique for spasticity. These are also discussed by Hsu et al., and the results of this network meta-analysis are thought to leave room for debate. There is currently no published research that details how to administer BoNT-A therapy to muscles that have fibrosis. Clinically, when spastic muscles are evaluated using sonoelastography, within the same muscle, there are scattered areas with severe fibrosis and areas with mild fibrosis. There is also room for debate as to whether injection into muscles with a high degree of fibrosis is effective or injection into muscles with mild fibrosis. Therefore, further research is needed regarding injection strategies for fibrotic muscles, rehabilitation strategies, and therapeutic applications of ESWT. Additionally, the need to investigate the synergistic effects of BoNT-A and ESWT, as well as its combination with other physical therapies, is essential.

## 4. BoNT-A for Lower Limb Spasticity and Walking Function

In this section, we have extracted previous systematic reviews regarding lower limb spasticity and walking ability. A study by Rosales et al., which extracted randomized controlled trials published between 1996 and 2004, reported on the efficacy and safety of BoNT-A. Among nine extracted studies, data on lower limb spasticity were available in two. The meta-analysis showed a significant difference in favour of BoNT-A compared with the placebo group in all RCTs targeting post-stroke spasticity [14]. Wu’s systematic review, specifically focusing on lower limb spasticity, identified seven RCTs. They reported significant improvements in muscle tone compared to that in a control group at 4 and 12 weeks after BoNT-A injection (4 weeks standardised mean difference [SMD] = 0.85, 95% confidence interval [CI]: 0.2–1.5; *p* = 0.001, 12 weeks SMD = 0.42, 95% CI: 0.07–0.77; *p* = 0.02) [37]. However, a recent systematic review by Doan et al. included 12 RCTs and reported a significant reduction in muscle tone in an intervention group compared to that in a control group at 4, 8, and 12 weeks after BoNT-A injections [38]. They also suggested optimal doses of 300 U of OnabotulinumtoxinA and 1000 U of AbobotulinumtoxinA for ankle spasticity.

From a different perspective, Foley conducted a systematic review using walking speed as an outcome measure [39]. They found an increase in walking speed of 0.044 m/s after intervention, with an effect size of 0.193 (95% CI: 0.033–0.353). In contrast, Doan’s systematic review, as mentioned earlier, reported improvement in walking speed at eight weeks after injection, but no significant difference compared to a control group [38].

Wu also reported on Fugl–Meyer Assessment (FMA) in their previous systematic review, indicating a significant improvement in the BoNT-A injection group compared to a control group (mean difference [MD] = 3.19, 95% CI: 0.22–6.16, *p* = 0.04). However, no significant difference was observed in walking speed [37].

Gupta focused on walking speed and quality of life in a systematic review of lower-limb spasticity [40]. Although they did not conduct a meta-analysis due to the small number of extracted papers (five RCTs), they found three papers with significant improvements in walking speed and two papers with significant improvements in FMA. Regarding the SF-36, one study was evaluated, but no significant improvement was observed compared with that in a control group.

It is particularly challenging to assess the effect BoNT-A therapy combined with lower-limb rehabilitation. Factors, such as the presence of orthoses, walking patterns, use of assistive devices, and patients’ perception of differences between the affected and unaffected sides during walking, make it difficult to prove effectiveness solely based on walking speed. The results of these reviews are summarised in Table 1.

## 5. The Effect of BoNT-A Combined with Rehabilitation for Lower Limb Spasticity on Walking Function

We extract and discuss studies regarding BoNT-A therapy and rehabilitation for lower limb spasticity, focusing on gait function. We conducted a literature search on BoNT-A and rehabilitation for post-stroke lower limb spasticity using PubMed, Scopus, CINAHL, Embase, PsycINFO, and CENTRAL databases until the end of December 2023, extracting only RCTs. Selected keywords included stroke, cerebral vascular accident, ischemic stroke, hemorrhagic stroke, botulinum toxin, botulinum toxin therapy, antispastic therapy, rehabilitation, physical therapy, occupational therapy, intensive rehabilitation, multidisciplinary rehabilitation, motor, function, ability, walk, and capacity. Keyword variations were set individually for each scientific database. References for all retrieved articles were reviewed to ensure that all relevant articles were included for data integration. As an example, the Pubmed search strategy is shown in Appendix A. For this purpose, search codes from a previous systematic review were consulted [19].

We assessed the methodological quality of selected studies as described in Cochrane reviews group [41]. A risk of bias table was developed, with a description and judgment (low risk of bias, high risk of bias, unclear risk of bias) of the following areas for each included study: (1) random sequence generation, which is selection bias (biased allocation to interventions) due to the inadequate generation of a randomized sequence; (2) allocation concealment, which is selection bias (biased allocation to interventions) due to the inadequate concealment of allocation prior to assignment; (3) the blinding of participants and personnel, which is performance bias due to knowledge of the allocated interventions by participants and personnel during the study; (4) the blinding of outcome assessment, which is detection bias due to knowledge of the allocated interventions by outcome assessors; (5) incomplete outcome data, which is attrition bias due to the amount, nature or handling of incomplete outcome data; (6) selective reporting, which reporting bias due to selective outcome reporting; and (7) other sources of bias, which are considered bias due to problems not covered elsewhere in the table. Two review authors independently performed quality assessment.

Any disagreements that arose among the authors were resolved through discussion or through the third author.

We found it difficult to conduct a meta-analysis because most trials on the effects of BoNT-A on stroke patients have focused on the effects of spasticity and few studies have focused on motor function.

Six papers focused on the combination of BoNT-A with conventional rehabilitation, while three studies explored additional therapies, including ES, 2 on robot-assisted rehabilitation, and 2 on taping and casting (Table 2 and Table 3) [6,42,43,44,45,46,47,48,49,50,51,52,53,54]. Except for one study, all the studies reported some improvement in walking ability compared to that in a control group.

Risk of Bias Assessment is shown in Table 4. A considerable proportion of the studies exhibit an high and unclear risk with regards to selection bias, primarily stemming from inadequate disclosure of the randomization process and allocation concealment [6,44,45,46,47,51,54,55,56,57]. Regarding allocation concealment, only four studies were deemed to have a low risk of bias [42,43,47,49,50,52,53]. Regarding blinding of participants and personnel, only 4 studies were deemed to have a low risk of bias [6,46,47,58]. This represents the difficulty of blinding in both BoNT-A treatment and rehabilitation. Regarding blinding of outcome assessment, 41% of studies were of low risk and appropriately described blinding of assessors [46,47,52,53,54,57,58]. Regarding incomplete outcome data, we judged 76% of studies to be of low risk [6,43,46,48,49,50,51,52,53,54,55,56,57,58]. In addition, Selective reporting showed Low risk at 64% [6,43,44,45,46,47,48,51,52,53,54,55,56], and Other bias showed Low risk at 58% [42,43,44,45,47,52,53,54,55,56].

BoNT-A has a dose dependent effect on reducing spasticity [13]. Therefore, the percentage of selected target muscles in the extracted studies is shown in Figure 1. The most selected streak was Gastrocnemius. It was followed by Soleus and Tibialis posterior. These muscles are involved in equinus varus of the ankle joint.

Among the extracted RCT studies, OnabotulinumtoxinA was used in six studies, AbobotulinumtoxinA in three studies, and IncobotulinumtoxinA in one study. In addition, the formulation name was unknown in three papers.

### 5.1. Rehabilitation Combination Therapies

Munari et al. compared regular treadmill training to backward treadmill training and found significant improvements in MAS and walking ability before and after the intervention in both the intervention and control groups, and in the intervention group compared to the control group. They reported that significant improvements in 10 MWT and static balance ability were observed [43]. Ding et al. added orthotic therapy to the combination of BoNT-A therapy and rehabilitation. The group with additional orthotic therapy showed improvements in Clinic Spasticity Influx, Berg balance scale, FMA, and Functional Independence Measure after three months, maintaining walking ability [45]. The timing and frequency of the evaluations varied across studies, making it challenging to form a consistent view. Therefore, the frequency and intensity of rehabilitation require further investigation.

### 5.2. Electrical Stimulation and Functional ES(FES) Combination

Three papers reported a combination of ES or FES (one duplicated study in different years) [47,48,49,50]. All studies compared before and after treatment and found some improvement in the intervention group. Baricich not only applied ES to the injected muscles but also to the antagonist muscle, the anterior tibialis [47].

### 5.3. Robot-Assisted Rehabilitation Combination

Two studies described the combination of robot-assisted rehabilitation [51,52]. Erbil et al. observed significant improvements in spasticity and walking function in both intervention and control groups, with additional improvements in static and dynamic balance in the intervention group [51]. In Picelli’s report, the intervention group showed significant improvements in both spasticity and 6-min Walk Test (6 MWT) [52]. Reports on rehabilitation using robotic technology in patients with stroke have increased in recent years, and the synergistic effects of combining neurorehabilitation strategies with BoNT-A suggest its potential for functional improvement.

### 5.4. Taping and Casting Combination

Two studies compared three groups: taping, casting, and stretching. Only the former showed significant improvement compared to that in a control group, and in an intervention group, improvements in spasticity and walking ability were maintained for an extended period [53,54]. Additionally, in the intervention group, improvements in spasticity and walking ability were maintained for an extended period compared to that the control group (stretching). This result is similar to that reported by Ding et al., who used orthotic therapy in combination with BoNT-A [45].

Among the extracted studies, many papers found significant improvement in evaluation items related to walking ability compared to the control group [42,43,44,45,46,47,48,51,52,53]. This result seems to indicate that combining BoNT-A therapy with rehabilitation may contribute not only to improving spasticity but also to improving walking function.

## 6. Evidence-Based Medicine on BoNT-A and Rehabilitation for Upper and Lower Limb Spasticity

We extracted papers related to injections to both the upper and lower limbs, which are often performed clinically. There is limited literature on the combined use of BoNT-A and rehabilitation for both upper and lower limbs, with only one RCT, one comparative study, and two prospective studies identified (Table 5) [55,56,57,58]. Prazeres et al. evaluated the combined effects of BoNT-A and rehabilitation on upper- and lower-limb function, with injections likely targeting the upper limbs (specifics not provided). They observed significant improvements in MAS, FMA, TUG, and 6 MWT three months after injection in an intervention group compared to that in a placebo-controlled group [58].

In a comparative study, Demetrios et al. investigated the effectiveness of purposeful training combined with upper- and lower-limb function training. The results showed no significant differences in upper limb and walking function improvements between the groups [57]. Amatya conducted a prospective study involving 35 participants, implementing both the BoNT-A and purposeful training in all cases [55]. They reported significant improvements in spasticity, Action Research Arm Test scores, and walking speed after intervention, with the improvement sustained for up to 12 weeks.

Hara et al. examined the effectiveness of a comprehensive rehabilitation program involving BoNT-A and training for both the upper and lower limbs [56]. The training protocol included six sessions of 20 min each, conducted daily over 12 days of hospitalisation. A specialised team of healthcare professionals, including physicians, physical therapists, occupational therapists, and nurses, designed and implemented a shared rehabilitation program based on patient assessments upon admission. The program incorporated various training elements, such as stretching, positioning, ROM exercises, ADL training, upper limb function training, walking exercises, balance training, and core control training. The results showed improvements in MAS score, joint ROM, and all other functional assessments. However, at the three-month follow-up, only improvements in the FRT were sustained. Subgroup analysis based on pre-intervention walking speed revealed that patients with a baseline walking speed of <0.4 m/s experienced the most significant improvement, suggesting a ceiling effect for patients with higher baseline walking speeds. This study suggests that patients with lower baseline walking speeds may derive maximum benefits from the combination of BoNT-A and rehabilitation, leading to substantial improvements in walking ability, and even enabling outdoor ambulation.

## 7. Effects of Repeated BoNT-A and Relationship with Rehabilitation

Given that the effects of BoNT-A are typically observed for a maximum of three months, re-administration is essential based on individual patient conditions after this period. Previous reports on repetitive treatments indicated continuous therapeutic effects and safety for spasticity; however, studies on their correlation with walking function and rehabilitation are limited [59,60]. Studies focusing on up to four repetitive BoNT-A injections for lower limb spasticity after stroke or traumatic brain injury have reported significant improvements in walking speed, stride length, and cadence at 12 weeks after the fourth injection compared to those at baseline [61]. Similarly, as in Hara’s study [56], when dividing cases into three groups based on walking speed and assessing them as pre- and post-intervention, cases with a barefoot walking speed of >0.8 m/s increased from 0% at baseline to approximately 20% at post-intervention. Hefter et al. also reported significant improvements with BoNT-A alone, particularly in the active ROM of the knee and ankle joints, sustained for up to one year from the start of administration [62].

Hara et al. focused on the effects of four BoNT-A injections combined with intensive inpatient rehabilitation [63]. The participants were 35 patients with chronic post-stroke upper and lower limb spasticity, 27 of whom used a brace at baseline. We observed significant improvements in MAS, 10 m walk test, TUG, and FRT scores from baseline to post-intervention. These improvements were sustained even after four injections. Importantly, not only did walking ability improve, but changes in brace requirements were also noted. Among orthotic users, 33.3% eventually discontinued brace use. Interestingly, all brace discontinuers exhibited a forward walking pattern, and the ankle joint ROM in these patients followed a trajectory similar to that of those who did not initially use a brace. This suggests that achieving a more normal walking pattern may ultimately contribute to improved walking function [63].

## 8. Walking Rehabilitation Strategies in Conjunction with BoNT-A

Based on previous reports, the combination of BoNT-A with rehabilitation contributes to improvements in walking function, including walking speed. Several key considerations are essential to maximise this potential (see Table 6). In addition, appropriate injection techniques based on thorough assessments are crucial, ultimately leading to an effective reduction in spasticity. Insights from previous studies on repetitive treatments suggest that early initiation of rehabilitation following stroke is vital for acquiring walking patterns that are tailored to individual cases. Although not all patients achieve a forward gait pattern during the early to subacute phases, obtaining this pattern as early as possible may advance treatment strategies for the chronic phase. Moreover, as indicated in our study and that by Esquenazi et al., patients with slow walking speeds at baseline may experience fewer ceiling effects, allowing for greater functional improvement [63]. Even if sufficient walking function is not achieved initially with BoNT-A administration, improvements in ankle joint ROM and modifications in orthotic use may lead to alterations in walking patterns, enhanced walking function, and importantly, gradual orthotic adjustments or the possibility of discontinuation [56,61]. Furthermore, neurorehabilitation methods, such as robot-assisted rehabilitation suggest synergistic effects in improving function.

Considering these key points and evidence, a treatment strategy for BoNT-A based on functional improvements was considered (Figure 2). Crucial evaluation indices included the degree of muscle fibrosis and walking speed. In cases of advanced fibrosis, where improvement in spasticity with BoNT-A alone may be limited, a treatment approach focusing on spasticity improvement using methods, such as ESWT is recommended. From a walking speed perspective, a multidisciplinary rehabilitation program tailored towards walking speed is recommended, as slower walking speeds are associated with a greater potential for improved walking function. Additionally, improvements in walking patterns may allow brace modification or discontinuation. In cases in which rehabilitation treatments demonstrate improvement, interventions, such as neurorehabilitation, including robot-assisted rehabilitation, may offer synergistic effects. Finally, in cases in which walking speed has improved to a certain extent, frequent BoNT-A administration is recommended to maintain function.

## 9. Limitation

In this section, we discuss the limitations of this study. With the goal of optimizing the effects of BoNT-A on lower limb spasticity to achieve functional improvement and gait reconstruction, this study aimed to extract and discuss the evidence obtained to date from multiple perspectives, ultimately finding several key points. In this regard, we mainly extracted RCT papers regarding gait function and gait reconstruction related to the combination of BoNT-A therapy and rehabilitation. However, it was difficult to extract data on walking ability that were common to each study. Some data used walking speed as an outcome. However, some studies did not adequately present the data. Therefore, a meta-analysis could not be performed. When focusing on walking ability, there is room for debate as to what constitutes improvement in walking ability. This is because walking speed does not equal walking improvement. As walking speed increases, the risk of falling also increases. Therefore, when walking, it is necessary to make a composite judgment based on multiple parameters such as balance evaluation and ankle joint range of motion. Second, the sample size of many papers is small. This represents one of the barriers in BoNT-A therapy research. Furthermore, from an RCT perspective, establishing a control group for BoNT-A administration is a clinically difficult issue. As a research strategy in general rehabilitation medicine, it is preferable to use rehabilitation alone as the control group, but in clinical practice, the accumulation of control groups is a barrier. Therefore, there is a limit to the number of patients that can be collected at one facility. In the future, it is necessary to consider multicentre research on the combination of BoNT-A therapy and rehabilitation, or multicentre international collaborative clinical research.

## 10. Conclusions

From the extracted studies, combining BoNT-A therapy with rehabilitation and adding other adjunctive therapies may contribute not only to improving spasticity but also to improving walking ability. We presented evidence on the effects of BoNT-A combined with rehabilitation established on evidence-based medicine for lower limb spasticity-related walking impairments. As emphasised, it is crucial to focus on certain indicators to maximise the effects of BoNT-A on spasticity and optimise the rehabilitation used in conjunction. By concentrating on these key considerations in daily clinical practice and building upon new studies, BoNT-A treatment for lower limb spasticity can be established as not only a treatment for spasticity but also as a neurorehabilitation strategy aimed at improving walking function.

## Figures and Tables

**Figure 1 toxins-16-00323-f001:**
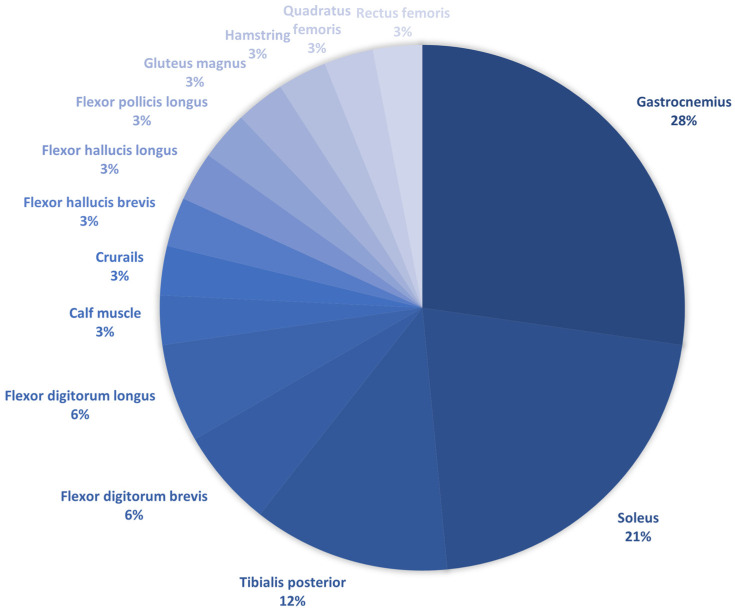
The percentage of selected target muscles in the extracted studies.

**Figure 2 toxins-16-00323-f002:**
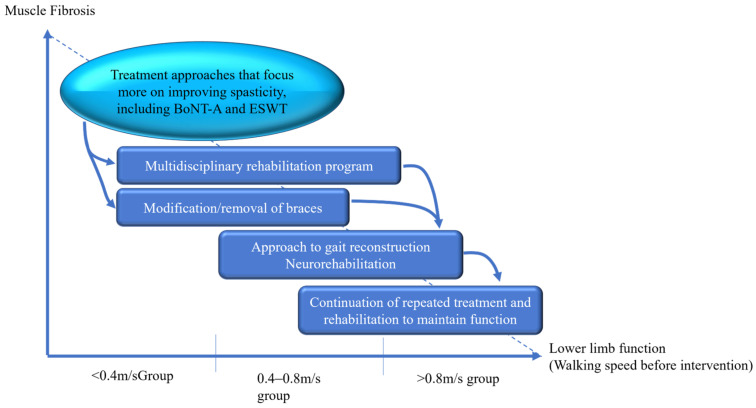
BoNT-A treatment strategy focused on functional improvement.

**Table 1 toxins-16-00323-t001:** Overview of a systematic review on lower limb spasticity and walking ability.

Study Name	Number of Extracted Papers and Design	Presence or Absence of Meta-Analysis	Muscle Tone	Walking Speed	Walking Ability	Other Evaluation Results
Doan et al., 2021 [38]	12 RCTs	+	Significant improvement compared to control group	No significant difference	N/A	N/A
Gupta et al., 2018 [40]	5 RCTs	−	Significant improvement in all extracted papers	There was a significant improvement in walking speed in three of the extracted papers	Two of the extracted papers showed significant improvement in FMA	No significant improvement in SF-36 in one of the extracted papers
Wu et al., 2016 [37]	7 RCTs	+	Significant improvement compared to control group	No significant difference	Significant improvement in FMA compared to control group	N/A
Foley et al., 2010 [39]	8 RCTs + Intervention study	+	N/A	Walking speed increased by 0.044 m/s before and after intervention	N/A	N/A
Resales et al., 2008 [14]	9 RCTs	+	Significant improvement compared to control group	N/A	N/A	GAS: Odds ratio = 5.85 (95% CI: 3.12–10.95)

+ indicates studies for which a meta-analysis was conducted. − denotes studies for which no meta-analysis was conducted. FMA, Fugl–Meyer assessment; GAS, Goal attainment scaling; N/A, Not Applicable; RCT, Randomized controlled trials; CI, confidence interval; SF-36, 36-Item Short Form Health Survey.

**Table 2 toxins-16-00323-t002:** BoNT-A and combined with rehabilitation for lower limbs.

Study Name	Study Design	Sample Size	Time between Onset and Treatment	BoNT-A-Dosage-Location	Rehabilitation Protocol	Assessments	Follow-Up	Results
Yu et al., 2023 [43]	RCT (vs. Routine Re)	I: 23 and C: 23	I: 3.4 (0.16) C: 4.2 (0.09) months	OnabotulinumtoxinA by ES 220–370 U QF, TP, G, FHL, FDB, and FHB	Routine physical therapy	MAS, FMA, 10 MWT and TUG	1, 4, and 12 weeks	The intervention group showed significant improvements in walking speed, TUG, and stride length compared to the control group.
Munari et al., 2020 [43]	RCT (backward treadmill training vs. forward treadmill training)	I: 7 and C: 11	I: 7 (3.4) C: 7 (3.7) years	OnabotulinumtoxinA by US 225 U—G, S	There was a significant improvement in walking speed in three of the extracted papers.	MAS, 10 MWT, static balance, and gait analysis	4 weeks	Significant improvements in MAS and walking ability were observed before and after the intervention in both the intervention and control groups. Significant improvements in 10 MWT and static balance ability were observed in the intervention group compared to the control group.
Roche et al., 2015 [44]	RCT (vs. BoNT-A only)	I: 19 and C: 16	I: 15.7 (6.9) and C: 7.3 (3.6) years	OnabotulinumtoxinA by ES —Gluteus magnus, RF, Crurails, Ham, S, Calf muscles, FDB, and FDL -Dosage had no data.	A standardized home-based self-rehabilitation program that consisted of three parts (10 min each). 4 weeks, Stretch, and task-oriented exercise.	MAS, the ABILOCO Scale, 10 MWT, 6 MWT, TUG, MRc, and Stairs test	1 month	Intervention group was significantly improved in gait speed, 6 MWT, and Stairs test
Ding et al., 2015 [45]	RCT (BoNT A + AFO + rehabilitation Vs. BoNT-A + rehabilitation vs. rehabilitation only)	I: 33 and C: 35	I: 17.0 (1.1) Observation: 16.4 (1.2) C: 15.4 (1.8) years?	BoNT-A by US -Dosage and location had no data.	Bobath concept, ROM, walking, massage, ADL training. Duration and frequency were not reported	CSI, FMA, BBS, and FIM	1, 3, and 6 months	Intervention and Observation group was significantly improved after one month in CSI, BBS, FMA, and FIM. Intervention group was significantly improved after 3 and 6 months compared other groups in CSI, BBS, FMA, FIM.
Tao et al., 2015 [46]	RCT (vs. placebo)	I: 11 and C: 12	I: 24.2 (12.2) and C: 23.2 (17.2) days	BoNT-A by ES 200 U—G, S, and TP	Gait training, the neurodevelopmental technique and motor relearning program physiotherapy (45 min every workday) and occupational therapy (30 min every workday).	MAS, FMA, 6 MWT, and modified BI	4 and 8 weeks	The gait analysis, FMA, and MBI results in Intervention group were better than those in control group.
Burbaud et al., 1996 [6]	Crossover RCT (vs. Placebo)	I: 10 and C: 13	I: 23.2 (36) and C: 23.8 (33) months	AbobotulinumtoxinA by EMG 1000 U—G, S, TP, and FDL	Active physiotherapy	MAS, FMA, and Gait speed	30, 90, and 120 days	Gait velocity was slightly but not significantly improved after BoNT-A injections.

FDB, flexor digitorum brevis; FDL, flexor digitorum longus; FHB, flexor hallucis brevis; FHL, flexor hallucis longus; G, gastrocnemius; Ham, hamstring; S, soleus; TB, tibialis posterior; QF, Quadratus femoris; RF, rectus femoris; I, intervention group; C, control group; AFO, ankle–foot orthosis; BBS, Berg balance scale; BI, Barthel Index; CSI, clinic spasticity influx; EMG, electromyography; ES, electrical stimulation; FMA, Fugl–Meyer Assessment; FIM, functional independence measure; FRT, Functional reach test; MAS, Modified Ashworth Scale; MRc, Medical research council scale; Re, Rehabilitation; ROM, joint range of motion; TUG, Timed Up and Go Test; US, ultrasound. 6 min walk test; 10 MWT, 10 m walk test.

**Table 3 toxins-16-00323-t003:** Study on adjunctive combination therapy for the lower limbs.

Study Name	Study Design	Sample Size	Time between Onset and Treatment	BoNT-A-Dosage-Location	Rehabilitation Protocol	Assessments	Follow-Up	Results
Combined ES, FES and Rehabilitation								
Baricich et al., 2019 [47]	RCT (BoNT-A + applied muscle stimulation + antagonist muscle stimulation vs. BoNT-A + applied muscle stimulation)	I: 15 and C: 15	I: 45.2 (51.9) C: 48.3 (39.1)	OnabotulinumtoxinA by US G (medial 50 U lateral 50 U, S (120 U)	I: ES to applied muscle and tibialis anterior muscle C: ES to the treated muscle 1 session 60 min for 5 consecutive days. Additionally, 60 min of physical therapy (strengthening, stretching, and walking training)	MAS, 10 MWT, ROM, MRC, and 2 MWT	10, 20, and 90 days	The intervention group showed significant improvements in walking speed, TUG, and stride length compared to the control group.
Fujita et al., 2018 [48]	Non-RCT (vs. BoNT-A only)	I: 17 (ES) and C: 17	I: 39.8 (37.7) and C: 75.2 (51.2) months	OnabotulinumtoxinA by US 300 U—G, S, TP, FDL, and FPL	Physical therapy was performed for 2 weeks (two 1-h sessions per day). Stretch, leg resistance exercises, low-frequency electrical stimulation, electromyographic feedback, walking exercises	MAS, Clonus score, ROM, and Gait speed	2 weeks	Gait speed changed significantly in the intervention group; in the group receiving BoNT-A + PT, biceps femoris activity and knee co-activation index during the loading response and tibialis anterior activity during the pre-swing phase increased after 2 weeks of intervention, while soleus and rectus femoris activity during the swing phase decreased. Soleus and rectus femoris activity during the swing phase decreased 2 weeks after the intervention.
Johnson et al., 2002, 2004 [49,50]	RCT (vs. rehabilitation only)	I: 10 (BoNT-A FES Rehabilitation) C:8 (Rehabilitation)	0–6 months: 9, 6–12 months: 9	AbobotulinumtoxinA by EMG 800 U—G, TP	A minimum of three sessions per week and outpatients two sessions per week	Walking speed, PCI, the Rivermead Motor Assessment, and SF-36	2, 4, 6, 8, and 12 weeks	Comparison of median walking speed (non-stimulated) in the control group with median stimulated walking speed shows a significant upward trend, with the trend lines being significantly different in location.
Combined Robot and rehabilitation								
Erbil et al., 2018 [51]	RCT (vs. BoNT-A + rehabilitation)	I: 32 (BoNT-A, RAT, and physical therapy) C: 16 (BoNT-A, physical therapy)	I: 39 (34.3) and C: 25.9 (24.6) months	BoNT-A by ES -Dosage and location had no data	30 min of RAT plus 60 min of physical therapy, whereas controls received 90 min of physical therapy for three weeks during weekdays	MAS, Tardieu Scale, TUG, BBS, and Rivermead Visual Gait Assessment	6 and 12 weeks	After treatment, there were significant improvements in spasticity, balance, and gait function in both the RAT and control groups. However, at post-treatment weeks 6 and 12, change from baseline TUG, BBS, and Rivermead Visual Gait Assessment were significantly higher in the RAT group than those in the control group.
Picelli et al., 2016 [52]	RCT (vs. BoNT-A only)	I: 11 (BoNT-A and RAGT) and C: 11 (BoNT-A)	I: 6.2 (4.2) and C: 6.1 (3.8) years	AbobotulinumtoxinA by US 750 U—G, S	RAGT (30 min a day for five consecutive days) Immediately after BoNT-A administration, all patients included in this study received a 60-min session of electrical stimulation of the injected muscles.	MAS, Tardieu Scale, 6 MWT	1 month	No difference was found between groups regarding MAS and the Tardieu scale measured at the affected ankle one month after BoNT-A. A significant difference in 6 MWT was noted between groups at the post-treatment evaluation.
Combined Taping, Casting and rehabilitation								
Carda et al., 2011 [53]	RCT (vs. rehabilitation only)	Taping: 24, Casting: 27, Stretching: 18	Taping: 46.9 (41.3), Casting: 52.3 (43.8), Stretching: 43.9 (39.6) months	IncobotulinumtoxinA by ES -each muscle 50–140 U -G, S	After the first week, all the patients, irrespective of the allocation arm, underwent 30 min of gait training and 20 min of plantar flexor muscle stretching each day for one week under the guidance of a senior physical therapist.	MAS, ROM, strength of ankle dorsal flexor muscles, 6 MWT, 10 MWT, and Functional Ambulation Categories	20 and 90 days	Intervention group showed better and longer lasting results than control group
Karadag-Saygi et al., 2010 [54]	RCT (vs. rehabilitation only)	I: 10 (BoNT-A, Kinesio taping) C:10	I: 35.2 (29) and C: 39.4 (30) months	OnabotulinumtoxinA by ES 150–200 U -G	Active-assistive range of motion and stretching exercises were given as a home exercise program to both groups. Exercises were assigned twice daily for 20 min for four weeks	MAS, ROM, Gait velocity, and step length	2 weeks and 1, 3, and 6 months	Improvement was recorded in both groups for all outcome variables. No significant difference was found between groups other than ROM, which was found to have increased more in control group at two weeks.

FDL, flexor digitorum longus; FPL, flexor pollicis longus; G, gastrocnemius; S, soleus; TP, tibialis posterior; I, intervention group; C, control group; 10 MWT, 10 m walk test; 2 MWT, 2 min walk test; 6 MWT, 6 min walk test; BBS, Berg balance scale; EMG, electromyography; ES, electrical stimulation; FES, functional electrical stimulation; MAS, Modified Ashworth Scale; MRc, medical research council scale; ROM, range of motion; TUG, Timed Up and Go Test; US, ultrasound; PCI, physiological cost index; RAGT, robot-assisted gait training; RAT, robot-assisted training.

**Table 4 toxins-16-00323-t004:** Risk of bias summary.

Risk of Bias	Random Sequence Generation	Allocation Concealment	Blinding of Participants and Personnel	Blinding of Outcome Assessment	Incomplete Outcome Data	Selective Reporting	Other Bias
Yu et al., 2023 [42]	Low	High	High	High	High	High	Low
Munari et al., 2020 [43]	Low	Low	High	High	Low	Low	Low
Roche et al., 2015 [44]	High	Unclear	High	High	Unclear	Low	Low
Ding et al., 2015 [45]	High	Unclear	High	High	Unclear	Low	High
Tao et al., 2015 [46]	Unclear	Unclear	Low	Low	Low	Low	Low
Burbaud et al., 1996 [6]	Unclear	Unclear	Low	Unclear	Low	High	High
Baricich et al., 2019 [47]	Low	Low	Low	Low	High	High	Low
Fujita et al., 2018 [48]	High	High	High	High	Low	Low	High
Johnson et al., 2002, 2004 [49,50]	Low	High	High	High	Low	High	High
Erbil et al., 2018 [51]	Unclear	High	High	High	Low	Low	High
Picelli et al., 2016 [52]	Low	Low	High	Low	Low	Low	Low
Carda et al., 2011 [53]	Low	High	High	Low	Low	Low	Low
Karadag-Saygi et al., 2010 [54]	Unclear	High	High	Low	Low	Low	Low
Amatya et al., 2019 [55]	High	High	High	High	Low	Low	Low
Prazeres et al., 2018 [58]	Low	Low	Low	Low	Low	Unclear	High
Hara et al., 2017 [56]	High	High	High	High	Low	Low	Low
Demetrios et al., 2014 [57]	High	High	High	Low	Low	High	High

**Table 5 toxins-16-00323-t005:** BoNT-A combined with rehabilitation for the upper and lower limbs.

Study Name	Study Design	Sample Size	Time between Onset and Treatment	BoNT-A-Dosage-Location	Rehabilitation Protocol	Assessments	Follow-Up	Results
Amatya et al., 2019 [55]	Prospective study	35	5.0 (4.0) years	IncobotulinumtoxinA by EMG and US Subscapularis, BB, FCU, FCR, FDS, FDP, G, S, TB, and FHL	Goal-oriented training tailored to each patient	MAS, ARAT, FAC, IPAQ, FIM, and EQ-5D	6 and 12 weeks	Significant improvements were observed in MAS, ARAT in the upper limbs, and gait/balance. Improvements were observed in the cadence and speed. Gait improvement was maintained until 12 weeks
Prazeres et al., 2018 [58]	RCT (vs. Placebo injection)	I: 20 C: 20	I: 34.15 (21.43) and C: 32.05 (14.89) months	AbobotulinumtoxinA -Dosage and location had no data	30 min, two times/week Stretching, mobilization, flexibility, endurance training, and functional training	MAS, FMA, 6 MWT, and TUG	3, 6, and 9 months	Significant improvements were observed in TUG and 6 MWT in both groups at three months after injection.
Hara et al., 2017 [56]	Prospective study	51	71 (62.7) months	OnabotulinumtoxinA Major muscles of the upper and lower limbs	1 session 20 min, 6 sessions/day, 12-day in-patient protocol. Comprehensive rehabilitation by setting goals and objectives based on evaluation at admission by physician, therapists, and nurses	MAS, ROM, FMA,10 MWT, FRT, and TUG	2 weeks and 3 months	Improvements in MAS, ROM, and all other motor function assessments were observed before and after the intervention. However, at three months of intervention, only FRT was maintained.
Demetrios et al., 2014 [57]	Comparative and controlled study (vs. rehabilitation)	I: 28 and C: 31	I: 2.3 (1.1–5.5) C: 2.5 (1.1–5.0) years	AbobotulinumtoxinA: 54 OnabotulinumtoxinA: 5 Major muscles of the upper and lower limbs	I: 3 or more times a week, 1 h session for about 10 weeks C: Less than twice a week, 1 h session Goal-oriented training tailored to each patient	MAS, ArmA, 10 MWT, and GAS	6, 12, and 24 weeks	Regarding MAS, significant improvement was observed at 6 and 12 weeks compared to the control. There was no significant difference in improvement between the two groups regarding upper limb function and gait. Both groups reached their goal at 24 weeks, and patient satisfaction was significantly high.

BB, biceps brachii; FCR, flexor carpi radialis; FCU, flexor carpi ulnaris; FDP, flexor digitorum profundus; FDS, flexor digitorum superficialis; G, gastrocnemius; S, soleus; TB, tibialis posterior; FHL, flexor digitorum longus; I, intervention group; C, control group; 6 MWT, 6 min walk test; 10 MWT, 10 m walk test; ARAT, Action Research Arm Test; ArmA, arm activity measure; EQ-5D, Euro-Quality of life; FAC, Functional Ambulation Classification; FIM, Functional Independence Measure; FMA, Fugl–Meyer Assessment; FRT, Functional Reach Test; GAS, goal attainment scaling; IPAQ, International Physical Activity Questionnaire; ROM, range of motion; TUG, Timed Up and Go.

**Table 6 toxins-16-00323-t006:** Key points in gait reconstruction strategy using botulinum toxin therapy combined with rehabilitation.

Identification of appropriate injection muscle and injection technique based on appropriate evaluation
Combined rehabilitation therapy aimed at improving function, including neurorehabilitation, etc.
Appropriate spasticity control
Improve ankle range of motion
Rehabilitation therapy that promotes forward gait pattern
Changing braces or turning off braces
Maintenance of efficacy by frequent BoNT-A administration

## Data Availability

Not applicable.

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
