# Peer review of "Gait Reconstruction Strategy Using Botulinum Toxin Therapy Combined with Rehabilitation"

_toxins, 2024, doi:10.3390/toxins16070323_

Round 1

Reviewer 1 Report (Previous Reviewer 4)

Comments and Suggestions for Authors

I thank the authors for considering the comments. The manuscript is substantially improved and can be accepted.

However, my recommendation is 'Minor Revision'. More detailed comments are given below.

Specific comments:

The article still contains gramatical errors.

Line 248: lower limb spas ticity? Please verify.

Line 257: Please verify.” We assessed the risk of bias in the included studies.”. It is correct?.

Lines 258-275. Please verify the sentence. This part seems like a response to the reviewers or a discussion between the authors.

Line 291: reducing spasticity[13]. Please verify.

Line 292: shown in a figure (Figure 1.)? Please verify.

Author Response

I thank the authors for considering the comments. The manuscript is substantially improved and can be accepted. 

However, my recommendation is 'Minor Revision'. More detailed comments are given below.

Specific comments:

→Thank you for your review. Thank you for checking carefully. We have made some corrections to the following comments. The corrections are shown in red.

The article still contains gramatical errors.

Line 248: lower limb spas ticity? Please verify.

→We removed the space. L251

Line 257: Please verify.” We assessed the risk of bias in the included studies.”. It is correct?.

→This has been deleted because it overlaps with the immediately following sentence. L260

Lines 258-275. Please verify the sentence. This part seems like a response to the reviewers or a discussion between the authors.

→Thank you for checking. The sentence you pointed out is a sentence about the paper extraction and selection of extracted papers in this paper. We added this sentence after referring to the previous review.

Line 291: reducing spasticity[13]. Please verify.

→Thank you for checking. We added a space. L294

Line 292: shown in a figure (Figure 1.)? Please verify.

→Thank you for pointing that out. We have made the changes as shown below.  L295

Therefore, the percentage of selected target muscles in the extracted studies is shown in Figure 1. 

Reviewer 2 Report (Previous Reviewer 5)

Comments and Suggestions for Authors

Dear authors,

thank you for the revision. The paper has improved.

Your rebuttal letter answering the questions and suggestions of the reviewer, is insuffucient and partly impolite: you say several times: "→We have corrected the sentence as it may have caused misunderstanding." To say clearly: There were no misunderstadings, there were inadaquancies.

I consider it a disrespect to the reviewer that the changes made are not visible in the rebuttal letter and the reviewer has to painstakingly look for the changes in the manuscript.

Minor: please give the ref numbers of the authors lines 141, 151, 207

Author Response

I consider it a disrespect to the reviewer that the changes made are not visible in the rebuttal letter and the reviewer has to painstakingly look for the changes in the manuscript.
→I don't understand what you mean. The editors commented that the corrections should be highlighted, and they are in red. Surely, if the corrections were difficult to see for you, you would have commented critically that they were difficult to see.

Minor: please give the ref numbers of the authors lines 141, 151, 207
→Thank you for pointing that out. I've added it. It's written in red. 

Reviewer 3 Report (New Reviewer)

Comments and Suggestions for Authors

Comments on the Quality of English Language

Author Response

Reviewer3

The subject and content of the study is very interesting.

However the aim of the manuscript and therefore the study design is not clear to me. Was this a systematic literature review? Narrative review? Meta-analysis? The structure of the manuscript is therefore not appropriate in my opinion.

→Thanks for your advice.

The important point of this paper is to extract not only the effect of the combination of BoNT-A therapy and rehabilitation for the lower limb, but also the key points that may be related to this effect. The key points are shown in Table 5. To reach this point, there are several points to be discussed, which are explained in the preceding paragraphs: the BoNT-A therapy, the combination with rehabilitation, the relationship with fibrosis, the effect of repeated administration, etc.

I do not understand why, for example, it is allowed in Europe and the U.S. to produce papers such as the following, but when similar forms are attempted to be issued from other countries, they are pointed out as not appropriate, etc.

Berube S, Hillis AE. Advances and Innovations in Aphasia Treatment Trials. stroke. 2019 Oct;50(10):2977-2984.

From p2-5, the authors use a narrative review to report on assessment for spasticity an muscle fibrosis to start with the actual literature review on p.6?

I advise the authors to follow reporting guidelines fort he chosen study design which can be found on the EQUATOR Network site : Reporting guidelines | EQUATOR Network (equator-network.org)

The purpose of this paper is stated at the end of the introduction as follows:

→This paper discusses the multifaceted aspects of spasticity assessment, administration, and rehabilitation with the goal of optimising the effects of BoNT-A on lower-limb spasticity and achieving functional improvement and gait reconstruction.

This is a repetition of my response to the previous reviewer,

The content of this paper does not have enough value compared to meta-analysis. We considered the possibility of meta-analysis during the planning stage of writing our paper. However, it was difficult to extract data on walking ability that was common to all papers. Some data used walking speed as an outcome. However, the presentation of data in some papers was insufficient. Therefore, we decided not to perform a meta-analysis on the papers extracted this time.

Including these, we have described the limitations of this research.

On the other hand, I believe that the keywords we discovered through this paper for the combination of BoNT-A therapy and rehabilitation have sufficient novelty. Therefore, we believe that this paper has sufficient value even if it does not reach a meta-analysis.

Additionally, to comply with standard reviews, we added the search words used to maintain reproducibility. We also added Risk of bias to evaluate the quality of each study. Therefore, the revised paper underwent the same steps as a standard review.

P5: BoNT-A for lower limb spasticity and walking function.

→ As pointed out, a line has been drawn through the deleted parts. L207

Line236 : this is confusing to me in this section, I suppose the effect of BoNT as stand alone therapy for walking function was addressed? So it is confusing to read that it is challenging to assess the effectiveness of lower limb rehabilitation. Rehabilitation includes more than BoNT injections to me.

→Lower limb rehabilitation was changed to BoNT-A therapy combined with lower-limb rehabilitation. L238

Especially since the combination of rehabilitation and BoNT on walking function is discussed in the next section 5: Evidence based medicine on BoNT-A and rehabilitation.

I would suggest to alter this title as the authors focus on gait function.

My suggestion is: effect of BoNT-A combined with rehabilitation for lower limb spasticity on walking function

→Thank you for pointing that out. I have changed the title as you suggested. L246

Section 7: Effects of repeated BoNT-A: So this means that all literature reported in section 6 were studies that evaluated the effect of only 1 single BoNT-A Intervention?

→That is correct. Please refer to the study design and follow up for each table.

P16: limitation: this concerns which part of the manuscript ? It is unclear to me if the limitations are about section 5-6-7-8? As the authors reported on systematic review as well as RCTs

→Thank you for pointing this out.

This is a limitation throughout this paper.

This is also due to the responses from the previous reviewers.

Some reviewers pointed out the methodology of this review, and one reviewer pointed out the small sample size of the extracted papers (we do not feel that clinically, but...).

Again, I don't understand why a review format that is accepted in the US and Europe is criticized as inappropriate when a similar document is issued in other countries.

We have revised the first half of the Limitation to the context as follows.

This is the part in red.

Please correct some grammatical and spelling

p.2 line 48,

→ BoNT type A →BoNT-A L48

p.3 line 143 we evaluated?,  

→Thank you for pointing that out. We have made the changes as shown below.  L143

In a study involving 102 patients with stroke, HMS of the gastrocnemius muscle was assessed by ultrasonography before BoNT-A injection and MAS scores of the knee and ankle joints were significantly reduced in all groups.

p6 line 248

→Thank you for pointing that out. We removed the space. L251

Round 2

Reviewer 2 Report (Previous Reviewer 5)

Comments and Suggestions for Authors

Dear authors, you did not really make a point to point reply in your rebuttal letter. This I must critizice. I have no firther suggestions.

This manuscript is a resubmission of an earlier submission. The following is a list of the peer review reports and author responses from that submission.

Round 1

Reviewer 1 Report

Comments and Suggestions for Authors

The study offers a comprehensive examination of the efficacy and potential advantages of botulinum toxin A (BoNT-A) therapy in the management of upper motor neuron syndrome, with a specific emphasis on reducing spasticity, enhancing joint range of motion, and alleviating pain. It underscores existing research gaps, particularly in addressing improvements in paralysis and functional enhancement. The incorporation of rehabilitation alongside BoNT-A treatment is advocated as a promising strategy to unveil latent active movement function obscured by spasticity, potentially enhancing overall motor function.

A critical aspect involves the determination of the targeted muscles. However, prior to acceptance, the paper requires refinement and a more detailed description of the utilization of botulinum neurotoxin in these patients. Specifically, a discussion on the intramuscular neural distribution of spasticity is warranted in the discussion section. Given that managing spasticity necessitates significant doses of botulinum toxin, accurately injecting multiple muscles with small doses is crucial.

The paper should reference and cite the following sources and discuss their implications in the discussion section:

"Intramuscular neural distribution of the serratus anterior muscle: regarding botulinum neurotoxin injection for treating myofascial pain syndrome"

"Anatomical Considerations for the Injection of Botulinum Neurotoxin in Shoulder and Arm Contouring"

"Elucidating intramuscular neural distribution of the quadratus lumborum muscle to propose an optimal trigger point injection for myofascial pain syndrome"

"Distribution of the intramuscular innervation of the triceps brachii: Clinical importance in the treatment of spasticity with botulinum neurotoxin"

In conclusion, the paper appears to offer valuable insights for readers.

Comments on the Quality of English Language

The paper seems to be great. 

Reviewer 2 Report

Comments and Suggestions for Authors

Dear Authors,

Your work explores a comprehensive strategy for reconstructing gait, uniting botulinum toxin A (BoNT-A) therapy with rehabilitation for individuals with lower limb spasticity. Its primary contributions lie in consolidating evidence on the critical aspects of improving paralysis and enhancing functionality through this integrated approach. The paper offers crucial insights into BoNT-A's effective management of spasticity, along with multidisciplinary rehabilitation programs. The review's strength lies in its thorough approach, providing a nuanced understanding of integrating BoNT-A and rehabilitation to not only address spasticity but strategically enhance walking function in neurorehabilitation.

However, it is essential to recognize a significant limitation inherent in the study type chosen for this work, specifically a review. While reviews play a crucial role in summarizing existing literature, the prevailing preference in the scientific community tends to lean towards more robust study designs that contribute novel insights. A standard review, although informative, may be perceived as relatively less robust from a scientific standpoint when compared to primary research studies or meta-analyses. Given the scientific community's inclination towards studies with stringent methodologies, it is recommended to consider fortifying the study design or integrating original research components to elevate the scientific merit of the manuscript, increasing its prospects for publication in the targeted journal.

Reviewer 3 Report

Comments and Suggestions for Authors

This manuscript reviewed the gait reconstruction strategy using botulinum toxin combined with rehabilitation. However, the authors missed some important methodology for reviewing those clinical studies and failed to provide statistics on those studies. For example, section 4 is a systematic review on BoNT-A for lower limb spasticity, but fails to provide the detailed methodology of the review and how to synthesize those research results, including the statistics. They also failed to comment on the sample size effects on the studies (most studies in this review have a very small sample size).

There are also issues with references, for example, Tables 2 and 3 did not provide references (used the alphabetic style for the study name, but very difficult to match the studies to the references listed in the reference section (numerical style).

There are some editorial issues (which may come from the template) that get cut off, for example, lines 67-68, line 262, just named a few, but there are quite a few throughout the manuscript.)

Comments on the Quality of English Language

Need moderate editing. 

Reviewer 4 Report

Comments and Suggestions for Authors

General comments

1. The way the document is presented is confusing. From the abstract to conclusion,  the objective of this review is not clear. During the reading progresses, it is observed that there is no order in the main document because of the absence of the general objective of this review. Besides, many grammatical errors make me lose interest in reading the manuscript. References and tables are not provided according to the publisher's instructions, among other things. The authors can improve this review by clearly indicating what the objective of conducting this review is.

2. The main problem is that the authors must justify the advantage of their concerning to previous studies and the new contributions of this manuscript with respect to others.

3. According to the title of this review, sections 5-8 comply with those that are in relation to the title provided. The other sections are outside the scope of the review.

4. Each section must include a short introduction and conclusion before describing all the results supporting it. This is to help the reader understand the importance of the results..

 Abstract

1. The abstract section must be modified. Indicate the main of this review.

2. Lines 21-23: The authors mention, "Based on these key points, the degree of muscle fibrosis and preintervention walking speed may serve as biomarkers for treatment strategies. Consider that these are not a biomarker. Please modify.

3. Line 24. As results? Remember that this is not an original article but a review article. 

4. The main document should describe the abbreviations to understand the readers better.

 Evaluation of Spasticity

5. The authors must add a table or figure indicating how to evaluate spasticity (MAS, MTS, ROM, etc).

 Muscle Fibrosis and Sonoelastography

6. Lines 152-156: Authors must modify the meaning of their paragraph. The way it is presented is a discussion, and this is not an original manuscript. The authors should remember that this is a review comparing the data. Besides, the appropriate references must be included.

7. Lines 163-170. Again, the authors mention the previous study performed by the working group. Please remember that it is a review. The form as presented is an original manuscript. Please modify it. Besides, introduce the appropriate references.

 Systematic Review on BoNT-A for Lower Limb Spasticity and Walking Function

8. The authors should include how many types of botulinum toxins are administered commercially in the botulinum toxin therapy combined with the rehabilitation.

9. The authors mention randomized controlled trials published between 1996 and 2004 that were extracted from Resales et al. ' Is this review based on another review?

The form, as shown in this section, is descriptive and focuses on the relationship between botulinum toxin therapy and rehabilitation. Please add a small conclusion about the benefit of the Systematic Review of BoNT-A presented in this section.

 Evidence-Based Medicine on BoNT-A and Rehabilitation for Lower Limb Spasticity.

10. Tables 2 and 3. Authors must describe their tables in the main document and provide references according to the publisher's standards.

 Evidence-Based Medicine on BoNT-A and Rehabilitation for Upper and Lower 

Limb Spasticity

11. Lines 261-263. The sentence needs to be completed. Please verify and correct

12. Lines 260-300. In these sections the authors only describe the results previously reported by the working gorup. However, for a better understanding of readers, it is necessary to add a comparison between each of the works and a discussion regarding the study's objective. 

13. Lines 317-332. In the previous paragraphs, the authors do not describe the works they refer to. However, in lines 317-332, they describe their works in detail and do not add the appropriate references to support this information—the same for section 7.

 14. The conclusion does not reflect the art of the work.

 15. Finally, the manuscript contains many typographical and grammatical errors. The authors should carefully review the entire main document.

Comments on the Quality of English Language

The manuscript contains many typographical and grammatical errors. The authors should carefully review the entire main document.

Author Response

The editor adjusted the font, resulting in words being separated in places we didn't intend.
I feel jealous of your lack of intelligence by commenting on that part more than necessary.
We should be aware of the incompetence of conducting peer reviews without sufficient knowledge.

Reviewer 5 Report

Comments and Suggestions for Authors

This paper reviews the multifaceted aspects of spasticity assessment, administration, and rehabilitation with the goal of optimising the effects of BoNT-A on lower-limb spasticity and achieving functional improvement and gait reconstruction. It is suggested that BoNT-A treatment for lower limb spasticity should be established not just as a treatment for spasticity but also as a therapeutic strategy in the field of neurorehabilitation aimed at improving walking function.

 General:

The paper, done by a prominent japanese group, contains a realy large amount of recent and actual information that is carefully edited and presented. However, there are still points to be clarified or improved - see in te following list.

Please check for spaces and punctuation in the manuscript.

Abstract:

Line 7f.: what do you want to say with this sentence? There are many papers covering improvement of paralysis and functional enhancement.

Introduction and other paragraphs:

Line 45: botulism, a rare but serious disease in animals -this is not really true – humans can also have botulism

Line: 78f: I am very astonished about this sentence – please explain including neuropathophysiology knowledge

Lines 142-156. You review an own study, but do not say something about rehabilitation, but summerise “indicating that even with the addition of rehabilitation, sufficient efficacy of BoNT-A may not be achieved in muscles with advanced fibrosis”

Line 160f: … found significant changes in correlation with MAS .. which ones?

Line 166f: … significant changes in the SWV were observed…  which ones?

Line 173: … In the context of addressing advanced fibrosis, Extracorporeal Shock Wave Therapy (ESWT) may be effective for treating advanced fibrosis. Extracorporeal Shock Wave Therapy has gained attention in recent years as a treatment for spasticity, and several hypotheses… after inroducing ESWT, please use it

Lines 176-180: please give references for 1) to 3)

Lines 189-191: Furthermore, the mechanisms of action of BoNT-A and ESWT differ, as demonstrated by studies showing no difference in efficacy between the motor and non-motor points of ESWT. I do not understand that what you want to say.

Line 191-192: you say that the second time, that “the mechanisms of action of BoNT-A and ESWT differ”– what shall I learn?

Line 228: you say “studies“ – I think they are reviews?

Line 229: is the Table caption correct?

Line 231: what is a trial’ ?

Line 250: please explain all used abbreviations of the Table 2 (QF, MWT, …)

Line 262: ..inter esting findings.. which ones?

Line 269: which ones are the “four) and what was the result?

Line 272 … improvements in the intervention groups before and after treatment .. who can improvements be measured before treatment?

Line 274: what was the result?

Line 283: Please give references to your statement

Line 290: please give the respective reference number

Line 296: say: Prazeres et al.

Line 368 Esquenazi et al., - please give the respective reference number

Line 392: Figure 1. BoNT-A treatment strategy focused on functional improvement. – Cancel 1 of the 2 identical lines.

Comments on the Quality of English Language

Please check the sentences listed above to clarify what you want to express.

Reviewer 6 Report

Comments and Suggestions for Authors

The paper’s purpose is to discuss the “multi-faceted aspects of spasticity assessment, administration, and rehabilitation with the goal of optimizing the effects of botulinum toxin A on lower limb spasticity and achieving functional improvement and gait reconstruction.”  The paper appears to try to present systematic reviews of the literature on several aspects relevant to the topic, including 1) botulinum toxin for lower limb spasticity and walking function, 2) botulinum toxin combined with rehabilitation for lower limbs, and 3) combined botulinum toxin A and rehabilitation for upper and lower limb spasticity.  However, none of these meet PRISMA guidelines for systematic reviews.  The conclusions reached often appear based on one or 2 selected publications and the authors’ own citations.  The level of information presented on various aspects of the paper varies from too detailed to insufficiently detailed.  For example, there is a lot of detail on spasticity and spasticity assessment, but too little information on sonoelastography and ESWT.

            There is inadequate information presented in the paper to support the recommendations in the figure and in Table 5. Some of the recommendations are not discussed in the paper itself.

It is suggested that the paper be re-focused as an evidence-based, systematic review of the relevant literature on the effect of combined botulinum toxin and rehabilitation on gait function in accordance with PRISMA guidelines.

Comments on the Quality of English Language

The paper needs moderate editing for English language and to correct typographical errors.